# Preliminary Study of the Relationship between Osteopontin and Relapsed Hodgkin’s Lymphoma

**DOI:** 10.3390/biomedicines12010031

**Published:** 2023-12-21

**Authors:** Valli De Re, Egesta Lopci, Giulia Brisotto, Caterina Elia, Lara Mussolin, Maurizio Mascarin, Emanuele Stefano Giovanni d’Amore

**Affiliations:** 1Immunopatologia e Biomarcatori Oncologici, CRO Aviano, National Cancer Institute, Istituto di Ricovero e Cura a Carattere Scientifico, IRCCS, 33081 Aviano, Italy; 2Nuclear Medicine Unit, IRCCS—Humanitas Research Hospital, Rozzano, 20089 Milano, Italy; 3AYA Oncology and Pediatric Radiotherapy Unit, CRO Aviano, National Cancer Institute, Istituto di Ricovero e Cura a Carattere Scientifico, IRCCS, 33081 Aviano, Italy; 4Pediatric Hemato-Oncology Unit, Department of Women’s and Children’s Health, University of Padua, 35122 Padua, Italy; 5Clinica di Oncoematologia Pediatrica, Azienda Ospedaliera—Università di Padova, 35128 Padova, Italy; 6Department of Pathology, San Bortolo Hospital, 36100 Vicenza, Italy

**Keywords:** *SPP1*, secreted phosphoprotein 1, *OPN*, osteopontin, HL, Hodgkin’s lymphoma, mo-DC, monocyte-derived dendritic cell, AYA, adolescents and young adults, PET, positron emission tomography, TLG, total lesion glycolysis

## Abstract

The primary objective of this study was to investigate the potential role of tissue osteopontin, also known as secreted phosphoprotein 1 (*SPP1*), as a contributing factor to an unfavorable prognosis in classical Hodgkin’s lymphoma (HL) patients who received the same treatment protocol. The study involved 44 patients aged 4–22 years, with a median follow-up period of 3 years. Patients with higher levels of *SPP1* were associated with tissue necrosis and inflammation, and there was a trend toward a poorer prognosis in this group. Before therapy, we found a correlation between positron emission tomography (PET) scans and logarithmic *SPP1* levels (*p* = 0.035). However, the addition of *SPP1* levels did not significantly enhance the predictive capacity of PET scans for recurrence or progression. Elevated SPP levels were associated with tissue mRNA counts of chemotactic and inflammatory chemokines, as well as specific monocyte/dendritic cell subtypes, defined by *IL-17RB, PLAUR, CXCL8, CD1A, CCL13, TREM1*, and *CCL24* markers. These findings contribute to a better understanding of the potential factors influencing the prognosis of HL patients and the potential role of *SPP1* in the disease. While the predictive accuracy of PET scans did not substantially improve during the study, the results underscore the complexity of HL and highlight the relationships between *SPP1* and other factors in the context of HL relapse.

## 1. Introduction

Hodgkin’s lymphoma (HL) is a disease that affects the lymph nodes and lymphatic system, and it can occur at any age. It accounts for 6% of childhood cancers, with a higher incidence in adolescents and young adults (AYAs) aged 15–25 years, compared to younger children [1]. The incidence of HL in Italy is the highest in Europe, with 64.6 cases per 1,000,000, as opposed to 29.7 cases per 1,000,000 elsewhere [2]. Fortunately, the disease is now highly treatable, with overall survival (OS) in young patients increasing in recent years, reaching 93.6% at 10 years under the LH2044 protocol [3]. Moreover, this figure is likely to improve further in the coming years with the introduction of new therapeutic strategies [4]. However, for patients who experience a relapse or do not respond well to the first treatment, achieving a cure can be challenging. Typical treatment consists of a combination of chemotherapy and, in certain cases, radiotherapy. In recent years, efforts have been made to reduce the use of radiation to minimize potential long-term side effects, such as the development of a secondary cancer or late effects on the heart, lung, and thyroid [5]. Determining the most suitable treatment for each patient requires a precise assessment of various negative prognostic factors, primarily including disease stage, bulk, systemic symptoms, and early response to chemotherapy. All of these factors are now considered in therapeutic group stratification (TL) within the EURONET-PHL-C2 adapted treatment protocol [6]. However, identifying the optimal prognostic factors remains challenging, as the choice of therapeutic strategies is constantly changing, and the number of cases with unfavorable outcomes is relatively small.

An important prognostic tool is metabolic imaging with fluorodeoxyglucose positron emission tomography (PET). This imaging technique provides information on bone marrow involvement independent of biopsy and compares various segmentation techniques for volumetric assessment at different stages: baseline (PET1), early (PET2), and late (PET3) assessment [7]. The technique also exploits the ‘Warburg effect’, which is observed in both tumor cells and inflammatory cells [8]. PET is employed to monitor treatment response after the second cycle of chemotherapy (early response assessment, or ERA). Following international standards, the Deauville five-point scale is considered the preferred visual method for distinguishing responses in patients with lymphoma [9]. Patients with an inadequate response may require added cycles of intensified therapy and a second PET scan to assess tumor response (late response assessment, LRA). Low uptake of the radiotracer fluorodeoxyglucose on PET at the time of diagnosis or in response to therapy is considered a good indicator of a favorable prognosis [6]. This assists in identifying low-risk patients who can reduce their therapy to avoid toxic side effects. However, metabolic imaging does not always identify all patients who are at risk of relapse, nor does it accurately distinguish between inflammatory processes and residual tumor disease. Therefore, given the range of methods available for volumetric analysis in HL beyond the classical bulk risk factor, which includes a unidimensional measurement of tumor mass [10], the AIEOP HL Study Group has developed a multifaceted prospective analysis focusing on volumetric analysis, such as SUVmax and SUVmean, as well as various methods like metabolic activity parameters, including metabolic tumor volume (MTV) and total lesion glycolysis (TLG), for patients participating in the EuroNet-PHL-C2 trial [11]. New biomarkers are also under investigation to enhance the precision of prognosis and deliver the most optimal treatment for children with HL. Ongoing research in this area is essential to further improve disease management and offer increasingly tailored and personalized therapies.

Identifying factors associated with an unfavorable outcome can be challenging, especially since many of them are interrelated (collinear). More advanced stage-IV HL, diagnosed when the tumor has spread beyond the lymphatic system to affect organs such as the liver, lung, bone, or bone marrow, is a well-recognized adverse prognostic factor.

Dysregulation of the glyco-phosphoprotein osteopontin (OPN)/*SPP1* has been implicated in various cancers [12], including specific types of lymphoma (e.g., large B-cell lymphoma, ALK-positive anaplastic large-cell lymphoma) [13,14,15]. *SPP1* has been found within the tumor microenvironment, where it has several roles, including attracting inflammatory cells and fibroblasts, promoting angiogenesis, protecting tumor cells from apoptosis, and enhancing the migratory and invasive ability of cells, thus facilitating the metastatic process [16]. *SPP1* undergoes secretion and experiences elevated levels of post-translational modification. The precise effects of these modifications on its function remain poorly understood, and differences in its functional response may also depend on tissue context [17,18]. Notably, thrombin has been shown to cleave *SPP1* and secrete a specific *SPP1* isoform (known as osteopontin-R), which contains domains for interacting with various integrin and CD44 receptors present on immune cells such as dendritic cells, mast cells, T cells, and neutrophils, thereby triggering an immune response in these cells [19]. Furthermore, through the use of alternative translation start sequences and splice mutations, *SPP1* exists in an intracellular form (iOPN) that is involved in several cellular functions such as migration, fusion, and proliferation [20,21,22,23]. In particular, the OPN-c variant has been implicated in breast tumor metastasis, where it appears to be unable to anchor to the extracellular matrix [24]. In addition, the intracellular form of *SPP1* has been detected in 100% of primary CNS tumors [25] and animal models have shown that it increases the invasive capacity of *Epstein–Barr* virus B cells across the nervous system, including the blood–brain barrier, in neurological diseases like multiple sclerosis [26]. Further, in vitro analysis has demonstrated that *SPP1* can also inhibit sensitivity to certain chemotherapeutic agents such as doxorubicin [27,28] and radiotherapy [29], implying that it might adversely impact tumor response. 

The main goal of this study was to assess whether *SPP1* is associated with adverse outcomes in adolescents, young adults, and children with HL and whether its combination with PET imaging could enhance the accuracy of patient prognosis. The second objective was to characterize factors and cell types within the HL group that are associated with elevated mRNA levels of *SPP1*.

## 2. Materials and Methods

### 2.1. Patient Characteristics

The study involved the prospective enrolment of newly diagnosed HL cases treated with the Italian multicenter response-adapted therapeutic EuroNet-PHL-C2 protocol (available online: https://clinicaltrials.gov/ct2/show/NCT02684708, last update 13 May 2021), with data collected from June 2017 to June 2020, and for which biological samples were available. The age of patients ranged from 4 to 22 years. The median follow-up period for assessing treatment response was 3 years. The study was approved by The Institute’s Ethics Committee (decision no. CRO-2016-12; 7 March 2016) and we also obtained approval from the ethics committees of the participating institutions in Italy. Written informed consent was obtained from patients or their legal guardians. 

The study included 44 tissue samples collected from 19 AIEOP centers. Patients who had received chemotherapy or radiotherapy prior to tissue collection and those with an insufficient sample for testing were excluded from the study. Clinical parameters including age, sex, clinical stage, bulky disease, B symptoms, extranodal infiltration, histological subtype, laboratory evaluation, and response to treatment were recorded. Response to treatment was evaluated through an interim assessment, conducted via 18F-fluorodeoxyglucose PET scan evaluation after 2 cycles of chemotherapy (early PET) or at a later assessment (late PET) following the EuroNet-PHL-C2 protocol [11].

### 2.2. Maximum Standardized Uptake Value

In HL patients, various segmentation techniques were employed for computerized PET tomography volumetric assessment at baseline [11]. This assessment included the determination of the maximum standardized uptake value (SUVmax), which represents the pixel with the highest uptake value, and the SUVmean, calculated as the mean value of uptake. Additionally, the semi-quantitative parameters obtained from these analyses included the metabolic tumor volume (MTV) and the total lesion glycolysis (TLG), where TLG is calculated as MTV multiplied by SUVmean. 

### 2.3. Immunohistochemistry 

A total of 33 tissue samples obtained from diagnostic HL biopsies underwent centralized investigation. Sections from lymph node or biopsy samples, following standard histology procedures (fixation with 4% buffered formalin and cutting into sections 3–5 μm thick), were subjected to heat-induced antigen retrieval at 95 °C in EDTA buffer pH9. Next, the endogenous peroxidase was blocked with hydrogen peroxide at ambient temperature for 5 min, and the samples were then incubated with the primary antibody against the antigen of interest for 30 min at ambient temperature. The primary antibodies used in this study, along with their clones and manufacturers, included CD15 (Carb-3, Dako, Milan, Italy ), CD20, (L26, Dako, Milan, Italy), CD3 (LN10, Leica Biosystem, Milan, Italy Novocastra, Leica Biosystem, Milan, Italy), CD30 (BerH2, Dako,, Milan, Italy), PAX 5 (24/PAX-5, BD Biosciences, DBA, Milan Italy), OCT 2 (OCT-207, Novocastra, Leica Biosystem, Milan, Italy), CD163 (10D6, Novocastra, Leica Biosystem, Milan, Italy), CD204 (SRA-E5, Cosmo Bio, DBA, Milan, Italy), CD11c (5D11, Monosan, DBA, Milan, Italy), and CD68r (PGM1, Dako, Mialn, Italy). The manufacturer’s recommended dilutions were followed for all the antibodies. Thereafter, the samples were washed and incubated with a secondary antibody (IgG, Dako) for 30 min. The signals were developed using Fast Red substrate (Abcam, Prodotti Gianni S.r.l., Milan, Italy), and Mayer’s hematoxylin (8 min) was used for counterstaining, applied for 8 min. To classify cases into different HL histotypes, a hematoxylin- and eosin-stained section was evaluated. Histological typing included nodular sclerosis (NS), mixed cellularity (MC), lymphocyte-depleted (DL), and lymphocyte-rich (LR). For the detection of viral RNA expressed during latent infection phases, the *Epstein–Barr* virus-encoded RNA (EBER) probe and in situ hybridization (ISH) technique was employed. The ISH protocol utilized differs from standard immunohistochemical analysis by the use of the EBER probe conjugated with fluorescein (ASR ISH5687-A, Leica) and the addition of an anti-fluorescein antibody. The Leica BOND automated system was employed, along with BOND Polymer Refine Detection, auxiliary reagents, and the ISH Protocol A staining protocol.

Photomicrographs were captured using a Leica microscope, Leica ICC50HD camera, and Leica Acquire software v.1.02.97 on a MacBook Pro 16′ 2.6 GHz 6-Core Intel Core i7 computer.

### 2.4. RNA Extraction and NanoString Quantification

The isolation of total RNA was performed on three 5 μm thick FFPE sections using the RNeasy DSP FFPE Kit (Qiagen, Hilden, Germany). To ensure the removal of contaminant genomic DNA, each RNA sample underwent DNase treatment. The RNA quality of the extracted RNA was assessed using the High Sensitivity RNA Screen Tape Kit (Agilent, Santa Clara, CA, USA) on the 2200 Tape Station System (Agilent, Santa Clara, CA, USA). The concentration of the extracted RNA was measured using the Qubit™ RNA HS Assay (Life Technologies Corporation, Eugene, OR, USA) on the Qubit Fluorometer 2.0 instrument (Invitrogen). Samples with RNA concentrations less than 40 ng/μL or absorbance A260/A280 ratios less than 1.5 were considered inadequate and therefore excluded from further analysis.

For RNA hybridization, 300 nanograms of RNA were incubated with probes of the nCounter PanCancer Immune Profiling Panel (NanoString Technologies, Seattle, WA, USA) overnight at 65 °C. After the RNA–probe complexes were bound and aligned in an nCounter Cartridge using the nCounter Prep Station, RNA counts were obtained by scanning 555 fields of view using the nCounter Digital Analyzer (NanoString Technologies, Seattle, WA, USA).

The raw data were normalized for each case by calculating the geometric mean of the positive controls and subtracting the background level, which was determined as the mean plus 2 standard deviations (2SD) of the counts of the housekeeping genes included in the assay. This was achieved using nSolver 4.0 software (NanoString Technologies, Seattle, WA, USA) with the Advanced Analysis module (2.0). The normalized data were then log2 transformed and fitted to a linear model for further analysis.

### 2.5. Statistical Analysis

Statistical analysis was performed using MedCalc statistical software, version 19.0.4 (MedCalc Software, Ostend, Belgium), and NSolver4 (NanoString). Descriptive statistics included conventional measures (mean, median, range). The association of factors with *SPP1* 25–75 percentiles was assessed using the chi-squared or Kruskal–Wallis test. Normalized log2-transformed mRNA expression data of cases were tested in univariate and multivariate regression analyses. Event-free time was defined as the time from the date of first treatment to the date of tumor relapse/progression or last follow-up. Multivariate analyses of expression data to predict the effect on event-free survival were performed using a Cox proportional hazards regression model. Event-free times were analyzed using the Kaplan–Meier method and compared using the log-rank test. A *p*-value < 0.05 was considered significant. 

## 3. Results

### 3.1. SPP1, Clinical Information, Laboratory Results, and Pathological Data

The study involved patients from 19 Italian centers. In Figure 1, the tissue *SPP1* levels of 44 HL samples are presented. The median for *SPP1* RNA counts was 325 counts, with the 25th and 75th percentiles at 181 and 926, respectively. To categorize *SPP1* expression levels, the quartile method was employed; levels exceeding the 75th percentile, which corresponded to 926 counts, were defined as *SPP1*^high^.

Clinical, laboratory, and pathological characteristics of the patients who participated in the study, categorized based on their *SPP1* expression levels as *SPP1*^High^ and *SPP1*^Low^, are reported in Table 1. Patients were treated according to their therapeutic stage groups and individuals in therapeutic group 1 did not undergo radiotherapy.

The study’s findings indicate that higher *SPP1* expression in HL patients was associated with more cases of necrosis and inflammation in the tumor microenvironment. This suggests that *SPP1* expression may be involved in processes related to tissue damage or death within the tumor and that *SPP1* may play a role in promoting or modulating inflammatory responses within the tumor. In addition, although not significant, we observed reduced levels of hematological albumin and hemoglobin (*p* = 0.052 and *p* = 0.073, respectively).

### 3.2. Higher SPP1 Count Associated with a Poor Prognosis

When comparing the event-free survival (EFS) curves of patients with *SPP1*^low^ and *SPP1*^high^, we observed a difference in mean survival duration, with 45.8 months for the *SPP1*^low^ group compared to 26.9 months for the *SPP1*^high^ group (Figure 2A). The Kaplan–Meier survival curves showed a hazard ratio of 2.6, with a 95% confidence interval ranging from 0.70 to 9.6; the *p*-value was calculated as 0.152. We also noted an association between elevated *SPP1* levels and a poorer response to treatment (Figure 2B), but the Jonckheere–Terpstra trend test results obtained were not significant, with a *p*-value of 0.19.

### 3.3. Positive Correlation between Value of PET Imaging and SPP1 

We compared the predictive value of various diagnostic positron emission tomography (PET) assessments using ^18^F-fluorodeoxyglucose (FDG) for tumor response. This analysis was performed using receiver operating characteristic (ROC) curve analysis (Figure 3). The results indicate that there was no significant difference observed between the different baseline scans. Among the assessed parameters, total lesion glycolysis (TLG) emerged as the most effective predictor, with an area under the curve (AUC) of 0.754. 

Furthermore, our study revealed a strong correlation between total lesion glycolysis (TLG) before treatment and the logarithmically transformed *SPP1* (log *SPP1*) levels, with a correlation coefficient (r) of 0.34 (*p* = 0.035) (Figure 4).

### 3.4. Comparing SPP1 to TLG for Predicting Tumor Relapse or Progression

Elevated *SPP1* expression might be linked to tumor recurrence or progression. ROC curve analysis suggested that tissue *SPP1* counts could serve as a potential prognostic biomarker for tumor recurrence or progression, with an area under the curve (AUC) of 0.645 when a cut-off of >1370 was used (Figure 5). When comparing *SPP1* to TLG, TLG appears to have the highest AUC of 0.754 for predicting the risk of recurrence/progression (Figure 5). Logistic regression analysis, using a forward selection method with a significance criterion of *p* < 0.05, was employed to build a prognostic model aimed at assessing whether a combination of *SPP1* > 1370 and TLG > 1178 could improve prognostic accuracy. However, the results indicate that only TLG > 1178 significantly improved the prognosis, with a *p*-value of 0.0127.

### 3.5. nCounter Immune Expression Analysis of the SPP1 Signature

To better understand the potential mechanism of tumor growth promotion by *SPP1*, a multigene expression analysis was performed on HL samples based on increased *SPP1* levels and relapsed/progressive disease. After Bonferroni *p*-value adjustment, it was found that seven genes were positively correlated with an increase in *SPP1* levels (logFC  >  1: *IL17RB, PLAUR, CXCL8/IL8, CD1A, CCL13, TREM1,* and *CCL24*) (Figure 6 and Table 2). 

Differentially expressed mRNA genes were found to be associated with cytokines linked with leukocyte chemotaxis (GO:0030595, fold change *p* < 0.0001, red color) and the proinflammatory and profibrotic pathway (WP5095, fold change *p* < 0.0002, green color), and regulation of M1 and M2 macrophage polarization (PMID: 28936211, fold change *p* < 0.001, blue color) as revealed by using STRING analysis (Figure 7).

### 3.6. SPP1 Signature Is Associated with Dendritic Immune Cell Types

Nanostring analysis using the nCounter PanCancer Immune Profiling Panel module revealed that *SPP1* expression positively correlated with dendritic cell type profiling (Figure 8A). As depicted in Figure 8B, an increase in DC cell profiling was also linked to a higher risk of cancer recurrence or progression.

To further characterize dendritic cell subtypes and their association with *SPP1* expression and patient outcomes, we utilized a customized gene signature. This profile encompassed plasmacytoid DC, conventional DC type 1, conventional DC type 2, and monocyte-derived DC, as detailed in Table 3. 

Figure 9 illustrates the results of the principal component analysis for the dendritic cell subset signature model across high, medium, and low *SPP1* expression categories. The analysis identified the DC subtype markers associated with the highest and lowest *SPP1* expression. Specifically, the mRNA levels of *CD163, CD1A, CD209, FCER1A, FCER1A, IL6, ITGAX,* and *NRP1* mRNA were found to exhibit a positive association with high *SPP1* rank levels. Conversely, *CX3CR1* mRNA levels were associated with a reduction in *SPP1* expression. 

Among the dendritic cell subset markers of the monocyte-derived dendritic cells, it is noteworthy that *CD1A* mRNA counts were also found to be associated with the risk of relapse and cancer progression, while *CD209* was associated only with an increase in *SPP1* levels. These associations were visually represented by the distribution in the violin plot shown in Figure 10.

## 4. Discussion

High *SPP1* expression has previously been shown to indicate poor prognosis in various types of tumors [16]. In the present study, we have presented for the first time novel preliminary insights demonstrating a correlation between high levels of *SPP1* mRNA expression and a decrease in event-free survival among pediatric and adolescent patients with classical HL. This is of particular interest as HL is relatively common among the young patient population and relapses are a particular challenge to treat. Enrollment of patients with a first diagnosis of HL and treatment with the same EURONET-PHL-C2 protocol minimizes treatment confounding factors. The collection of laboratory, demographic, clinical, and histological information from the same patients, in addition to the molecular results, is another advantage of the study, which can provide a large amount of data from the same patients in a short period of time, further reducing the potential for confounding. Our research has emphasized the correlation between high *SPP1* counts and well-known negative histological prognostic factors in cancer, including tissue necrosis and inflammation. Additionally, we observed a positive trend in the relationship between *SPP1* expression and the tumor’s response to EURONET-PHL-C2 treatment, a phase II treatment focused on adapting treatment strategies to minimize the use of radiotherapy in an effort to lower potential late toxicity experienced in patients during the follow-up period [6]. 

One function of *SPP1* that has been suggested in the literature to be associated with poor prognosis is the promotion of the metastatic process. However, in our series, we did not find an association between patients having an extranodal tumor and an increase in SSP1 (Table 1). Conversely, we found a correlation with increased mRNA expression of cytokines associated with leukocyte infiltration and proinflammatory and fibrotic pathways after Bonferroni adjustment (*p* < 0.0001, Figure 6 and Figure 7). Most of these proteins have been reported to play important roles in inflammation and cancer, which implies a potential contribution to adverse events in HL. Below, we list the potential implications of these genes based on the existing literature.

*IL17RB* is codifying a receptor expressed on the epithelial cells of various organs and lymphocytes. It binds to the cytokines IL17B and IL17E. When stimulated by IL17E, IL17RB can activate NF-kB and induce the production of IL-8 [30]. These findings are significant in the context of HL since (i) *IL-8*, which can have implications for HL progression, is among the significantly upregulated genes listed in our series; (ii) *NF-kB* signaling is known to be a crucial factor in HL pathogenesis; and iii) silencing *IL17RB* expression with siRNA has been shown to enhance the anticancer activity of doxorubicin, potentially enhancing the efficacy of existing treatments [31]. 

The urokinase plasminogen activator receptor (uPAR), encoded by the *PLAUR* gene, acts as a receptor for urokinase plasminogen activator (uPA), an enzyme involved in the activation of plasminogen to plasmin, the active form. uPAR is responsible for localizing and promoting plasmin formation around cells that expose this receptor. This process has significant implications depending on the environment, thereby influencing either local thrombolysis or extracellular matrix degradation. This proteolytic cascade, which involves uPAR and plasmin, has been linked to vascular disease and cancer progression and metastasis [32]. Notably, our previous proteomic studies have already shown correlations between the coagulation process and increased components of the plasminogen activation system correlated with an increased risk of recurrence in HL [33,34,35].

CD1A is an antigen-presenting protein mainly expressed on immature dendritic cells, which play a crucial role in innate immunity. CD1A can bind both self and non-self lipid and glycolipid antigens and presents these antigens to T-cell receptors found on natural killer lymphocytes and T lymphocytes, providing crucial assistance in detecting and responding to foreign invaders, including cancer cells.

CCL13, also known as MCP-4 (monocyte chemoattractant protein 4), is a ligand for several receptors (CCR1, CCR2, CCR3, CCR5, and CCR11), and it has a significant impact on the immune response and HL microenvironment. Its primary function is to recruit monocytes, immature DCs, eosinophils, T cells, and NKs to the site of inflammation [36]. CCL13 plays a crucial role in the development of M2 tumor-associated macrophages (TAMs), a subset of macrophages often associated with immunosuppression, tumor angiogenesis, and tumor spread. Notably, high plasma levels of *CCL13* mRNA have been associated with slow early response in children with HL and are considered a potential risk stratification marker in adults with HL [37,38].

CCL24 (Eotaxin-2) has shown promise as a biomarker in multiple cancer types, including colon cancer, non-small-cell cancer, and nasopharyngeal carcinoma. CCL24′s plasma levels have been found to increase in cancer and metastasis. It is primarily produced by M2 macrophages, which are often associated with tumor-promoting activities, including immunosuppression and angiogenesis [39]. In clear-cell renal cancer carcinoma, CCL24 binding to the CCR3 receptor was found to create a positive autocrine loop that predicts prognosis [40]. In non-small-cell lung tumors, plasma CCL24 levels have been significantly correlated with diagnostic PET imaging, both during and after radiation therapy [41]. Therefore, while the specific role of CCL24 in HL has not been mentioned, the broader understanding of its potential as a biomarker in various cancers and its associations with disease progression and prognosis suggest that it could be a valuable area of investigation for HL as well.

Triggering receptor expressed on myeloid cells 1 (TREM1) is a transmembrane protein of the immunoglobulin (Ig) superfamily constitutively expressed on the surface of peripheral blood monocytes and neutrophils. It is upregulated by Toll-like receptor (TLR) ligands, which lead to increased production of pro-inflammatory cytokines and chemokines by neutrophils and macrophages. This, in turn, promotes their migration, thereby enhancing the inflammatory immune response in the tumor microenvironment [42]. TREM is associated with shorter survival in patients with solid tumors [43], suggesting that it could be an area of interest for research in the context of HL.

Further research is needed to explore these potential implications in greater detail and to uncover the specific roles of the above-listed proteins in the context of HL and its adverse events.

The correlation observed in our study between higher levels of *SPP1* and the infiltration of dendritic cells in the tumor microenvironment was significant. This was demonstrated through mRNA cell profiling and further characterized the dendritic cell subsets in the context of HL (Table 3). The presence of dendritic cells in the tumor microenvironment suggests their potential role in the immune response against HL. Dendritic cells are known for presenting antigens to T cells, which is a crucial step in initiating an immune response. The increased dendritic cell subpopulation derived from monocytes recruited to inflamed tissues [44] has a role in antigen presentation to T cells and polarization of the T cell response, rather than being a proliferative stimulator like conventional DCs [45]. Moreover, it is known that CD1a+ monocyte-derived dendritic cells are the primary source of lipid antigen presentation to T cells and of T-helper-cell polarization [46,47]. The specific T-helper cells around tumor cells in HL underscores their importance in influencing the adaptive immune response to the disease. CD209 binds to specific glycans and is associated with dendritic cell migration [48]. Interestingly, the migratory capacity of CD209+/CD14+ dendritic cells has been shown to be significantly inhibited by a *JAK/STAT* pathway inhibitor (tofacitinib) used in the treatment of various autoimmune diseases [49]. The *JAK/STAT* pathway is known to be constitutively activated in HL, suggesting that it may be involved in recruiting dendritic cells to the site of inflammation. These findings suggest a complex interplay between dendritic cells, antigen presentation, and the immune response to HL. 

The overall results of the present study suggest an association between high *SPP1* levels and a worse prognosis in HL. Currently, a widely used prognostic marker in HL is pre-treatment PET imaging. There are several measurement methods to evaluate PET imaging, and in our series, total metabolic glycolysis of the lesions resulted in the best approach (Figure 3). When comparing the combination of *SPP1* levels and PET with PET results alone, the addition of *SPP1* levels did not lead to an improvement in the prediction of patient classification for treatment risk assessment. In fact, the samples identified as possibly having a worse prognosis, defined as having high *SPP1* levels, were all already included in the samples identified by PET and therefore did not add any new prognostic information. Further research and a larger sample size may be necessary to ascertain the full implications of *SPP1* levels and their potential role in disease relapse or progression in the context of classical HL.

## 5. Conclusions

In conclusion, while these findings are preliminary, they collectively shed light on the association between the infiltration of monocyte-derived dendritic cell subtypes, the increase in *SPP1* mRNA expression in HL tissue samples, and patient outcomes. These results provide valuable insights for understanding the underlying pathological mechanisms in classical HL, specifically in children, adolescents, and young adults. However, despite the significance of these findings, the increase in *SPP1* levels has not demonstrated a substantial enough contribution to improve the predictive efficacy of PET imaging as an independent indicator of relapse or progressive disease prior to treatment with the EURO-NET-PHL-C2 protocol. Further research and a larger sample size may be necessary to ascertain the full implications of *SPP1* levels and their potential role in disease relapse or progression in the context of classical HL. These preliminary findings lay the groundwork for future investigations into *SPP1*’s complex role and the immune response in HL.

## Figures and Tables

**Figure 1 biomedicines-12-00031-f001:**
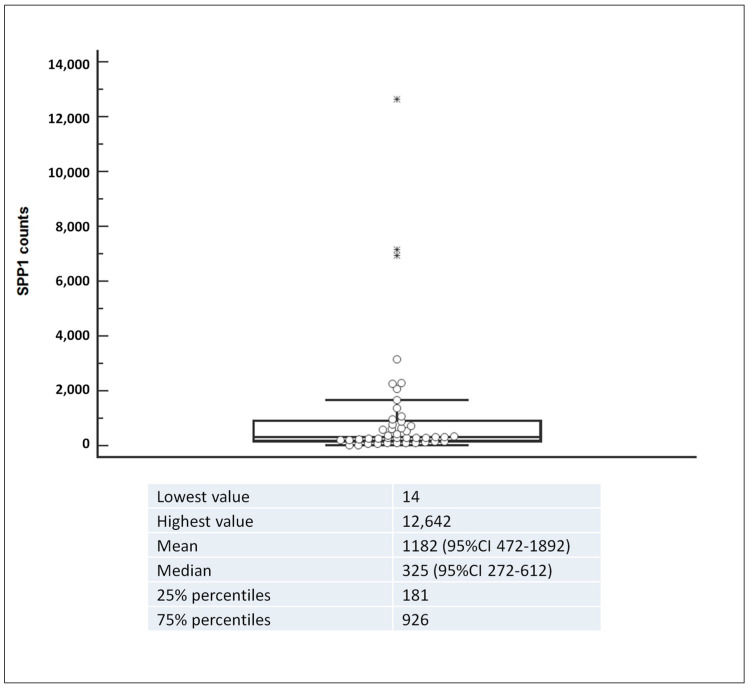
Box and whisker plots showing the level of *SPP1* mRNA expression in tumor samples from 44 HL patients.

**Figure 2 biomedicines-12-00031-f002:**
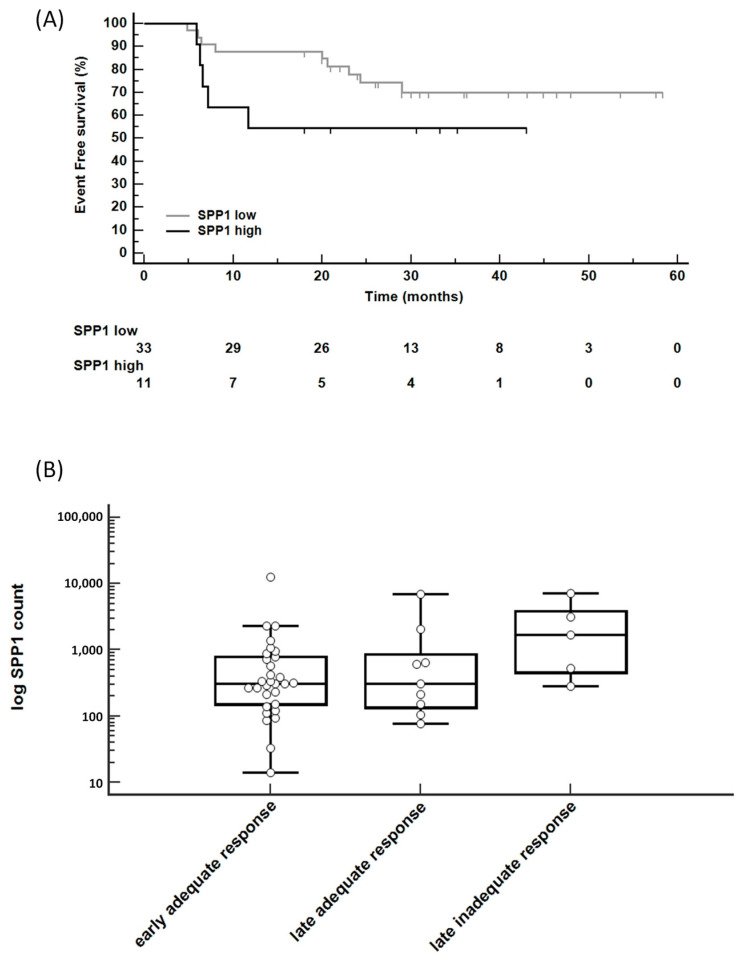
(**A**) Kaplan–Meier survival curves to assess event-free survival in patients with *SPP1*^low^ and *SPP1*^high^ mRNA levels. (**B**) *SPP1* level increased in cases with a late, inadequate response to treatment.

**Figure 3 biomedicines-12-00031-f003:**
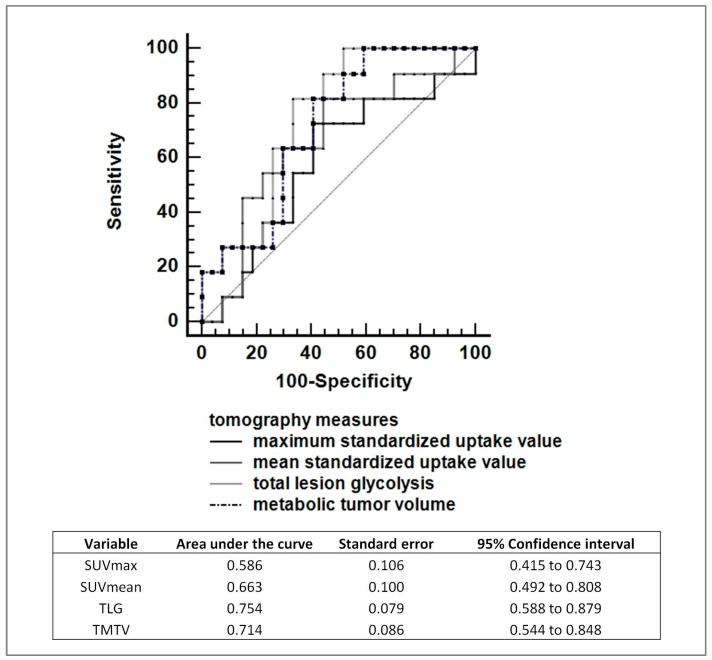
The graph displays receiver operating characteristic curves for various PET tomography parameters used to predict disease recurrence or progression over a median 3 years of follow-up [11].

**Figure 4 biomedicines-12-00031-f004:**
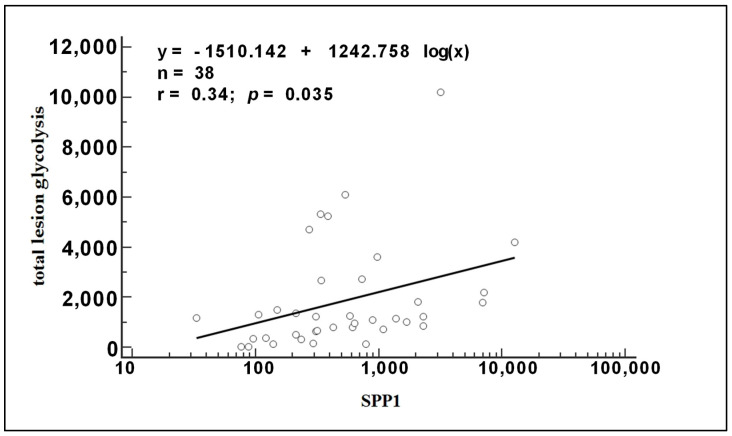
Regression analysis showing a correlation between each log *SPP1* mRNA level and the total lesion glycolysis assessed in the scan.

**Figure 5 biomedicines-12-00031-f005:**
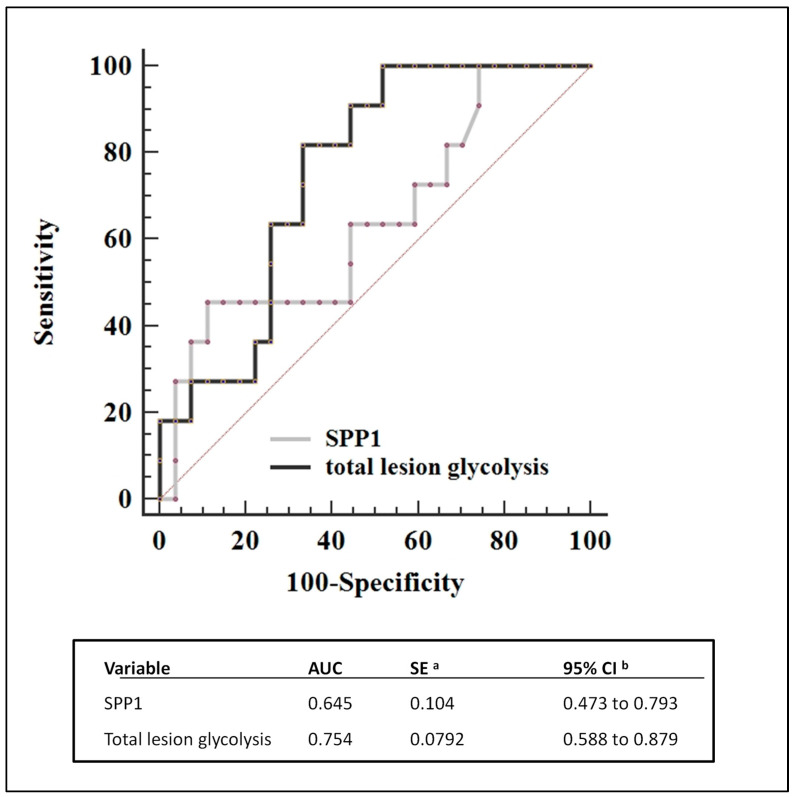
The scan that assesses total lesion glycolysis (TLG) has the highest area under the curve value, indicating its superior performance over *SPP1* in predicting the risk of recurrence or progression. ^a,^ standard errors, ^b^ 95% confidence intervals.

**Figure 6 biomedicines-12-00031-f006:**
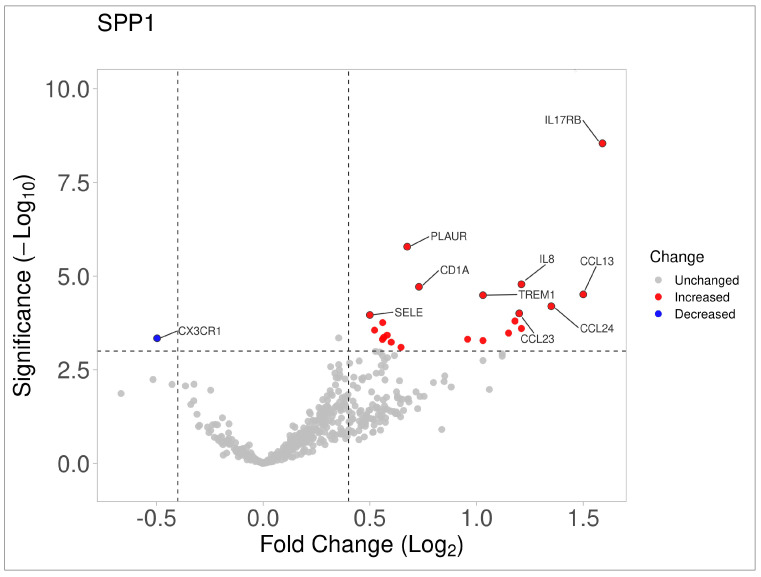
Volcano plot showing differential immune gene expression according to high *SPP1* mRNA expression levels and recurrence covariate.

**Figure 7 biomedicines-12-00031-f007:**
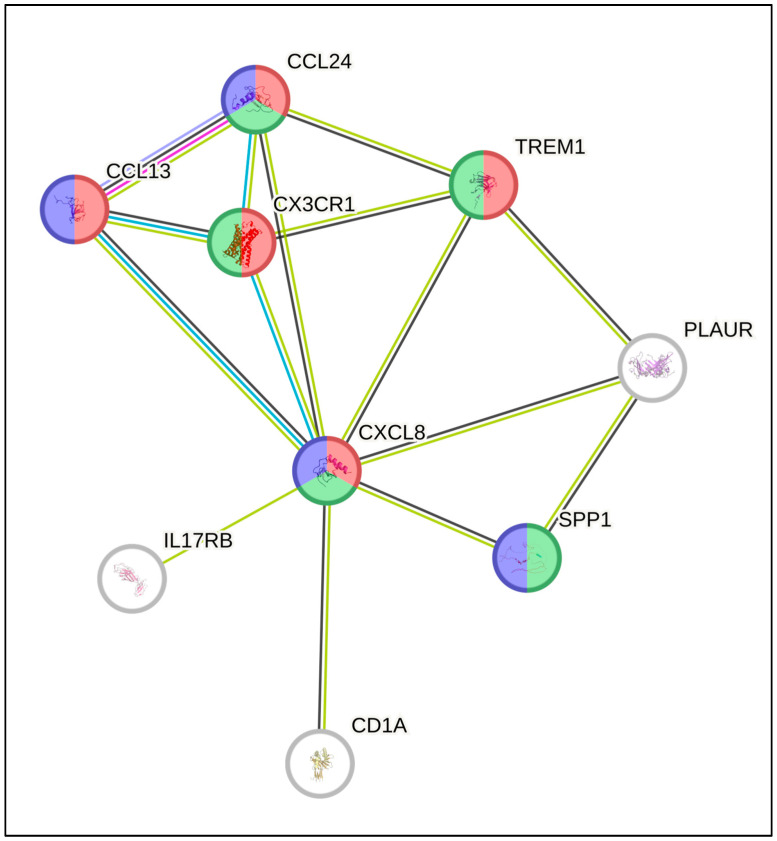
Molecular pathways associated with higher *SPP1* expression, along with genes identified as significant by using Bonferroni-adjusted volcano plot analysis (String online tools). Pathway enrichment analyses were involved in cytokines associated with leukocyte chemotaxis (red color), pro-inflammatory and profibrotic pathways (green color), and regulation of M1 and M2 macrophage polarization (blue color).

**Figure 8 biomedicines-12-00031-f008:**
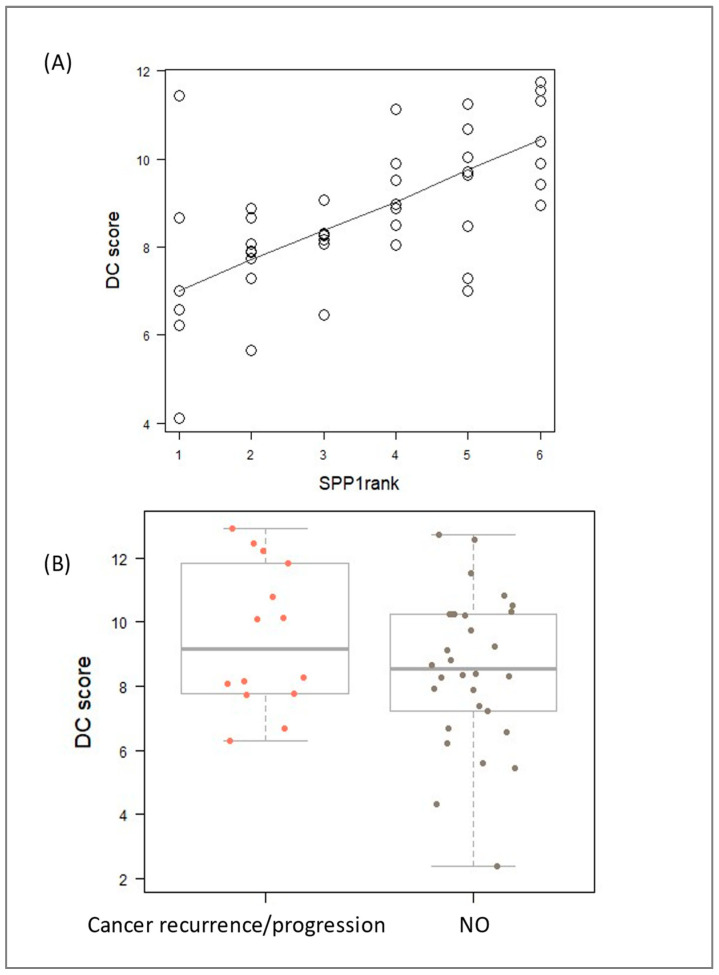
Cluster analysis of genes revealed a positive correlation between observed mRNA profile of dendritic cells (DC score) and both the increasing *SPP1* rank level (**A**) and the risk of tumor relapse or recurrence (**B**).

**Figure 9 biomedicines-12-00031-f009:**
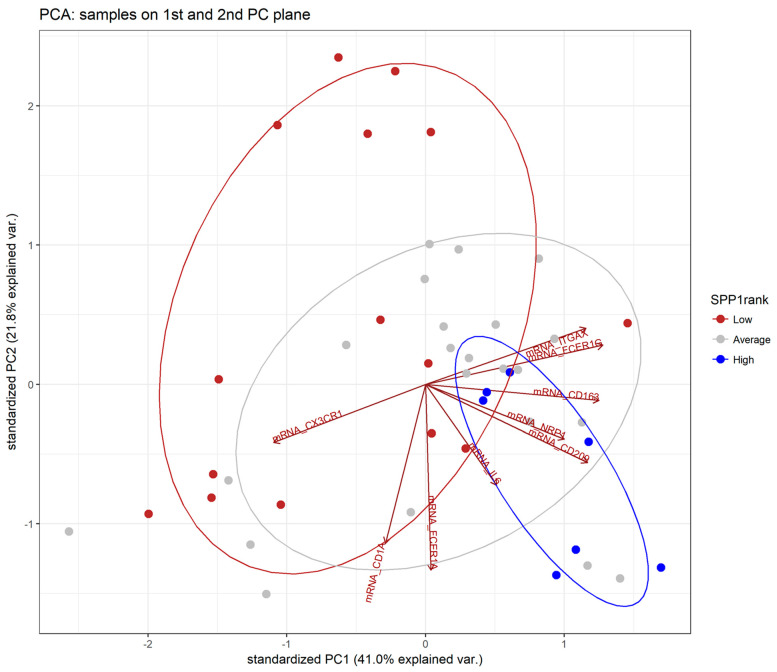
Principal component analysis for the dendritic cell subset signature model across high, average, and low *SPP1* expression levels.

**Figure 10 biomedicines-12-00031-f010:**
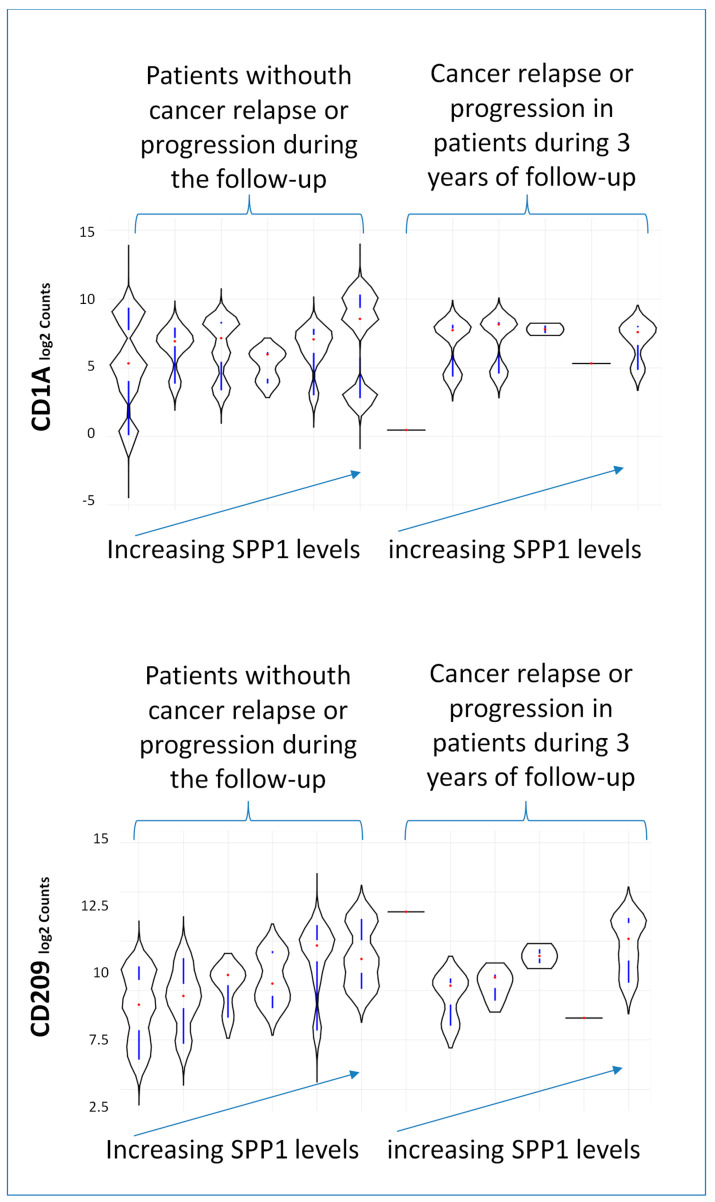
Violin plot illustrating *CD1A* and *CD209* expression within patient groups based on both increasing *SPP1* rank and the risk of relapse or tumor progression.

**Table 1 biomedicines-12-00031-t001:** Clinical, laboratory, and histological data at diagnosis of HL study cohort according to *SPP1*^High^ and *SPP1*^Low^ levels. Levels >75th percentile were defined as *SPP1*^high^.

	*SPP1* ^Low^	*SPP1* ^High^	*p*-Value *
** Clinical data **			
Age (mean)	15	14	0.358
Sex (M/F)	15/18	5/6	1.000
Stage (I-II/III-IV)	7/26	3/8	0.681
B-symptom (A/B)	10/23	3/8	0.850
Bulky (yes/no) 1nv	16/16	5/6	0.797
Extranodal disease (yes/no)	20/13	5/6	0.385
Pleura (yes/no) 26 nv	7/6	4/1	0.322
Lung (yes/no) 26 nv	9/4	3/2	0.718
Pericardic (yes/no) 26 nv	0/13	0/5	--
Liver (yes/no) 26 nv	2/11	0/5	0.366
Bone (yes/no) 26 nv	6/7	0/5	0.114
bone marrow (yes/no) 26 nv	5/8	2/3	0.954
Therapeutic group (1–2/3)	9/24	2/9	0.580
Relapse/Progression (yes/no)	9/24	5/6	0.268
Early response (adequate response/inadequate response or progression)	24/9	6/5	0.268
Late response (adequate response/inadequate response or progression)	7/2	2/3	0.265
Deauville (>2/≤2)	4/5	4/1	0.301
** Laboratory data **			
Erythrocyte sedimentation rate (median)	84	19	0.109
Albumin (median)	3.8	3.3	0.052
Alkaline phosphatase (median) 25 nv	127	124	0.351
C-reactive protein (median)	11	9	0.949
Ferritin (median) 10 nv	256	335	0.792
Hemoglobin (median) 6 nv	10.7	9.5	0.073
Immunoglobulin A (median)	261	201	0.298
Immunoglobulin G (median)	1320	1462	0.412
Immunoglobulin M (median)	101	136	0.891
Lactate dehydrogenase (median) 6 nv	360	264	0.786
Lymphocytes	1.4	1.2	0.851
White blood cell	10	13	0.160
Neutrophils	10	9	0.866
Platelets	358	437	0.427
Fibrinogen 5 nv	477	574	0.228
Protein	7.4	7.7	0.355
** Histological data **			
Histotype (Nodular Sclerosis/Mixed cellularity) 2 others	20/6	11/7	0.720
Necrosis (yes/no) 14 nv	2/19	7/2	**0.042**
Inflammation (yes/no) 14 nv	4/17	5/4	**0.049**
Grading (1/2) 14 nv	5/16	5/4	0.119
CD15 (+/−) 25 nv	8/2	7/2	0.386
CD20 (+/−) 4 nv	5/24	2/9	0.945
CD3 (+/−) 15 nv	0	0	---
CD30 (+/−) 4nv	29	11	---
PAX5 (+/−) 4 nv	26/3	8/3	0.186
OCT2 (+/−) 2 nv	4/16	1/8	0.565
CD163 (+/−) 12 nv	7/16	6/3	0.147
CD204 (+/−) 16 nv	1/19	1/7	0.494
CD11c/CD163 (−1/1) 15 nv	16/4	6/3	0.642
CD68r_7% 15 nv	10/10	8/1	0.096
*Epstein–Barr* virus (positive/negative)	8/25	4/7	0.441

nv, data not recorded; *, Fisher’s exact test or Mann–Whitney test.

**Table 2 biomedicines-12-00031-t002:** Table summarizing upregulated and downregulated genes according to higher *SPP1* mRNA expression levels.

Gene	Gene Description	Log2 FC	Lower CL	Upper CL	*p*-Value	Bonferroni Adjusted *p*-Value
*IL17RB*	Interleukin-17 receptor B	1.59	1.18	2	2.86 × 10^−9^	1.73 × 10^−6^
*CCL13*	C-C motif chemokine 13	1.5	0.872	2.13	3.08 × 10^−5^	0.0186
*CCL24*	C-C motif chemokine 24	1.35	0.758	1.95	6.38 × 10^−5^	0.0386
*IL8*	Interleukin-8	1.21	0.722	1.69	1.65 × 10^−5^	0.00998
*F13A1*	Coagulation factor XIII A chain	1.21	0.618	1.8	0.00025	0.151
*CCL23*	C-C Motif Chemokine Ligand 23	1.2	0.654	1.74	9.90 × 10^−5^	0.0599
*CXCL6*	C-X-C Motif Chemokine Ligand 6	1.18	0.624	1.74	0.000158	0.0957
*CCL26*	C-C Motif Chemokine Ligand 26	1.15	0.573	1.72	0.000331	0.2
*CCL11*	C-C Motif Chemokine Ligand 11	1.12	0.491	1.75	0.00119	0.719
*FN1*	Fibronectin 1	1.12	0.482	1.76	0.00137	0.829
*TREM1*	Triggering receptor expressed on myeloid cells	1.03	0.6	1.47	3.23 × 10^−5^	0.0195
*MME*	Membrane Metalloendopeptidase	1.03	0.492	1.56	0.000525	0.318
*CD209*	C-type lectin receptor	0.958	0.463	1.45	0.000484	0.293
*CD1A*	T-cell surface glycoprotein CD1a	0.73	0.435	1.03	1.93 × 10^−5^	0.0117
*PLAUR*	Urokinase plasminogen activator surface receptor	0.675	0.438	0.911	1.64 × 10^−6^	0.000995
*IL1RN*	Interleukin 1 Receptor Antagonist	0.646	0.297	0.995	0.000794	0.48
*COL3A1*	Collagen Type III Alpha 1 Chain	0.615	0.265	0.965	0.00132	0.8
*CD1E*	T-Cell Surface Glycoprotein CD1e	0.6	0.286	0.915	0.00058	0.351
*ITGB3*	Integrin Subunit Beta 3	0.582	0.247	0.917	0.00151	0.915
*OSM*	Oncostatin M	0.581	0.288	0.874	0.000377	0.228
*TNFRSF8*	TNF Receptor Superfamily Member 8	0.565	0.276	0.855	0.000433	0.262
*PTGS2*	Prostaglandin-endoperoxide synthase 2	0.56	0.294	0.826	0.000174	0.105
*CXCL3*	Chemokine (C-X-C motif) ligand 3	0.559	0.27	0.848	0.00049	0.296
*IL1R2*	Interleukin 1 Receptor Type 2	0.559	0.241	0.877	0.00132	0.801
*TNFRSF12A*	TNF Receptor Superfamily Member 12A	0.547	0.243	0.851	0.00107	0.65
*LILRA5*	Leukocyte Immunoglobulin-Like Receptor A5	0.527	0.235	0.819	0.00102	0.614
*PLAU*	Plasminogen Activator, Urokinase	0.522	0.265	0.779	0.000277	0.168
*SELE*	Selectin E	0.5	0.272	0.729	0.000109	0.0661
*CD276*	Cluster of Differentiation 276	0.354	0.172	0.536	0.000447	0.271
*CX3CR1*	CX3C chemokine receptor 1	−0.497	−0.753	−0.241	0.000457	0.277

FC, fold change; CL, confidence limit (log2).

**Table 3 biomedicines-12-00031-t003:** mRNA profiles used to characterize dendritic cell subsets.

Plasmacytoid DC		Conventional DC Type1		Conventional DC Type2		Monocyte-Derived DC	
*CCR7*	C-C chemokine receptor type 7	*CD8A*	T-cell surface glycoprotein CD8 alpha chain	*CD14*	Monocyte differentiation antigen CD14	*CD14*	Monocyte differentiation antigen CD14
*CD209*	CD209 antigen, Pathogen-recognition receptor	*ITGAE*	Integrin alpha-E heavy chain	*CD163*	Scavenger receptor cysteine-rich type 1 protein M130	*CD1A*	T-cell surface glycoprotein CD1a
*CLEC4C*	C-type lectin domain family 4 member C	*ITGAX*	Integrin alpha-X	*ITGAM*	Integrin alpha-M	*CD1C*	T-cell surface glycoprotein CD1c
*NRP1*	Neuropilin-1	*THBD*	Thrombomodulin	*CX3CR1*	CX3C chemokine receptor 1	*CD209*	CD209 antigen; Pathogen-recognition receptor
*SIGLEC*	Sialic acid-binding immunoglobulin-like lectin	*XCR1*	Chemokine XC receptor 1	*CD1C*	T-cell surface glycoprotein CD1c	*ITAGX*	Integrin alpha-X
*IFNA1*	Interferon alpha-1	*BATF*	Basic leucine zipper transcriptional factor ATF-like	*CD2*	T-cell surface antigen CD2	*ITGAM*	Integrin alpha-M
*CD4*	T-cell surface glycoprotein CD4	*IRF8*	Interferon regulatory factor 8	*ITAGX*	Integrin alpha-X	*MRC1*	Macrophage mannose receptor 1
*IRF7*	Interferon regulatory factor 7	*IRF4*	Interferon regulatory factor 4	*CD33*	Myeloid cell surface antigen CD33	*IRF4*	Interferon regulatory factor 4
*IRF8*	Interferon regulatory factor 8	*CD33*	Myeloid cell surface antigen CD33	*FCERI*	High-affinity immunoglobulin epsilon receptor subunit alpha	*FCGR1*	High-affinity immunoglobulin gamma Fc receptor I
*TLR7*	Toll-like receptor 7	*CX3CR1*	CX3C chemokine receptor 1			*FCER1G*	High-affinity immunoglobulin epsilon receptor subunit gamma
*TRL9*	Toll-like receptor 9					*IL23*	Interleukin-23 receptor
*IFNA*	Interferon alpha					*TNF*	Tumor necrosis factor
*IL6*	Interleukin-6						
*TNF*	Tumor necrosis factor						

## Data Availability

All data are included in the text.

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
