# Peer review of "Preliminary Study of the Relationship between Osteopontin and Relapsed Hodgkin’s Lymphoma"

_biomedicines, 2023, doi:10.3390/biomedicines12010031_

Round 1

Reviewer 1 Report

Comments and Suggestions for Authors

The study by De Re et al. analyzes SPP1 (osteopontin) RNA expression in a cohort of samples from young Hodgkin lymphoma patients, correlating it with clinical and analytical parameters. They also correlated SPP1 with chemokines RNA expression using NanoString technology. The authors also analyze the association of SPP1 expression with PET-FDG, with cHL relapse.

The topic is interesting, and the manuscript is generally well written (although it would benefit from an English speaker revision). However, there are some considerations that the authors should take into account.

Major.

The number of samples/patients is too low to reach significant conclusions. This reviewer understands the difficulty of collecting samples from young HL patients, but some statistical analyses are overestimated and the study is underpowered. For example, they divided the series into three groups according to SPP1 expression. The criteria seem sound, but with only data from 44 samples, the groups are too small for the statistical analysis to be robust. Additionally, they did a multivariate analysis including eight variables, which should not include more than three variables at most. The subsequent analyses are weak due to this. The absence of internal and/or external validation is another shortcoming of the analysis.

A statistician should supervise all the analyses.

nCounter is a robust platform to study gene expression in paraffin samples, and the information the authors extract from these data is certainly of interest. However, they should expose more clearly the meaning of these findings. Some figures are included but without a clear explanation of the aim of the representation. For example, they described what Figures 9 and 10 represent but not the results or conclusions derived from them: are the results significant?.

Minor

In section 3.4, line 333, the authors state: “The results showed that patients at risk of relapse or progression were well identified by the combination of albumin, SUVmean, haemoglobin and SPP1”. But, in the table in figure 5A, SPP1 is not significant (HR: 1, P = 0.08)

In line 224, It is written (HR 3.4172, hazard ratio SPP1 > 926 counts compared to SPP1 < 181), but they did not include the p-value.

Some figure legends need to be translated into English (for example, “si”/no), and some legend is missing (Figure 5, for example).  

Comments on the Quality of English Language

The English language is generally good, although I detected some inaccurate expressions. It would benefit from an English speaker revision.

Author Response

We thank the reviewer for its observation. As observed by reviewer himself it is difficulty to obtained a large number of samples due to the low age of patients and the low number of patients with a relapsed Hodgkin’s lymphoma.  Accordantly with the reviewer suggestion we reduce the number of SPP1 percentile groups from 3 to 2, given the preliminary analysis of the study.  Table 1 and Figure 2A have been changed accordingly. The comparison of relapse survival times between SPP1 levels and diagnostic PET imaging was now limited to TLG imaging, and we excluded SPP1 levels from the model in the multivariate analysis because SPP1 only approaches to statistical significance (P=0.08), and then a larger number of cases needs to be tested to draw conclusions about the potential contribution of SPP1 to the TLG data as a predictor for relapse risk in HL. In the discussion section, this limitation of the study was highlighted. With the exception of this single study point, all study results were unchanged from the previous submission.

Our nanostring analysis data suggest that some pathways involved in leukocyte infiltration and inflammation are associated with increased SPP1 levels. Since SPP1 levels were more frequent in patients with tumour relapse, these same pathways may be indirectly involved in the process of HL relapse. Taken together, most of these proteins have been reported to play important roles in inflammation and cancer, suggesting a likely contribution to adverse events in HL. Possible implications of these genes from the literature are listed in the discussion.

As suggested, part of Figure 9 and Figure 10 have been removed as not essential to the scope of the study.

Legend to figure 5 had been included

The figure legends have been reviewed for English

Reviewer 2 Report

Comments and Suggestions for Authors

The manuscript is on mRNA transcriptomic signature of osteopontin in children and adolescents with Hodgkin lymphoma and its association with relapse. Unfortunately, the manuscript is hard to read and follow due to changing formatting and style. e.g. the title is too long, there are too many keywords and it is not clear from the beginning that the data are prelimianry of nature. I suggest to report the findings following established scientific writing guidelines, also caring for consistent writing, e.g. Hodgkin's lymphoma vs. Hodgkin lymphoma. The conclusion is very vague and short, without mentioning practical and theoretical implications of the findings. The figure legends are very long and should be shortened, especially as the differentiation between legend and text is not always possible.

Comments on the Quality of English Language

Extensive editing of English language and format required. Advice from an English native speaker is recommended due to many typos.

Author Response

Accordanlty with the reviewer’s suggestions we shortened the title, there omitted some keywords and underlined that data are prelimianry of nature.

We revised the manuscript for scientific writing guidelines and typos.

The conclusion was extended. Practical and theoretical implications of the findings were reported.

Figure legends have been revised. They are now shorter and more clearly distingued from the text.

Reviewer 3 Report

Comments and Suggestions for Authors

The article “ mRNA transcriptomic signature of osteopontin in children and  adolescents with Hodgkin lymphoma and its association with relapse” is very interesting and I have some comments to make to improve the article.

-     Add the meaning of EBV+

-     Clarify whether the laboratory tests were performed with the same methodology in the different hospitals where the patients were recruited.

-     Add the commercial brands of the different immunohistochemical analyzes and the methodology of section 2.3

-     Name the specific markers of dendritic cells, the Jonckheere-Terpstra analyses, ROC curve, AUC, PCA Bonferoni P value adjusted, (statistical analysis) and the programs used to prepare the figuresIt would be interesting to know the age range of the patients.

-     Correct minimal grammatical errors

Thank you very much

Author Response

We have eliminated some of the abbreviations and given the meaning.

We confirm that the laboratory tests were carried out using the same methodology in the different hospitals where the patients were recruited.

We have detailed the immunohistochemistry protocol section by adding the codes and manufacturers of the monoclonal antibodies used, as requested

Age range of patients has been added. Specific markers of dendritic cells have been reported in Table 3, statistical test analysis and the programs used to generate the figures have been reported in the figure and in the statistical methods section.

Grammatical errors have been corrected.

Round 2

Reviewer 1 Report

Comments and Suggestions for Authors

De Ra et al.'s new version of the manuscript has addressed my major concerns. However, still some minor revisions are needed.

The conclusions regarding the clinical significance of SSP1 expression should be "smoothed". Some relations indeed show a trend, but they are not significant and, therefore, not conclusive.

Figure quality should be improved, mainly Figure 6 (the gene names are superimposed and, therefore, impossible to read).

In section 3.2. the said: "only marginally significant p-value of 0.152". However, a p-value of 0.152 is not significant by far. This should be corrected.

Something similar in Figure 2 legend: "The statistical significance achieved was only marginal (p=0.19)." 

Author Response

Dear reviewer

Many thanks for your accurate review, all changes had been highlited as a colored revison in the text and in the pdf version

We revised both the abstract and the discussion section to smoothed the significance of SPP1 expression.

The resolution of Figure 6 has been adjusted using GraphPad Prism software.

We rephrased version of the sentence in the text “The Kaplan-Meier survival curves showed a hazard ratio 2.6, with 95% confidence interval ranging from 0.70 to 9.6, the p-value was calculated as 0.152”.

We revised the legend of figure 2 as “"However, it is worth noting that the statistical significance obtained was not significant, with a p-value of 0.19."

Reviewer 2 Report

Comments and Suggestions for Authors

I thank the authors for revising their manuscript. Unfortunately, there are some points that have not been adressed sufficiently yet. Still, it is not clear from the beginning (titel etc.) that the data are preliminary of nature. Also, the figure legends are still very long and should be shortened.

I would recommend to rewrite the abstract, as it is not very clear. Also, the importance of EURONET-PHL-C2 is unclear.

Comments on the Quality of English Language

Moderate editing of English language required

Author Response

dear Reviewer,

Many thanks for your accurate review, all changes had been highlited as a colored revison in the text and in the pdf version

We have edited the title and added phrases that emphasize the preliminary nature of the study in the abstract, and discussion.

We have rewrite the abscract and  briefly described the importance of the EURO-NET trial in the context of the study (reporting again the reference 11 also in the discussion and not only in the introduction).

English editing that has been done

Round 3

Reviewer 2 Report

Comments and Suggestions for Authors

I thank the authors for extensively editing their manuscript. However, besides the in general low overall significance of the findings from a small prelimianry investiagation, some of the major issues raised in the previous review rounds are still pending. E.g. the tables and figures are difficult to understand due to the many unexplained abbreviations. Also, please ensure that the resolution of the figures is high enough for reading.For Fig. 5, it is unclear whether it has two separate parts or not.

It is still unclear, what the practical and theoretical implications of the findings could be as they are very vague such as ""Further research and a larger sample size may be necessary to ascertain the full implications of SPP1 levels and their potential role in disease relapse or progression in the context of classical HL."

Comments on the Quality of English Language

Extensive editing of English language and format required. Advice from an English native speaker is recommended. It is also good to remove unnecessary terms such as beginning the conclusion with: In conclusion...

Author Response

the tables and figures are difficult to understand due to the many unexplained abbreviations.

> Most of the abbreviations found in Tables and Figures correspond to the conventional nomenclature for genes, commonly used to present data from tools such as nanostring, volcano plot ecc, as seen in numerous published papers. However, since requested, we have reported the full names of all genes present in tables and figures, while avoiding the use of abbreviations in the text.

Also, please ensure that the resolution of the figures is high enough for reading.

> Figures are all in an adequate resolution; please, check those in .tiff, which have been included in a separate file as the usual paper submition of MDPI.

For Fig. 5, it is unclear whether it has two separate parts or not.

> Regarding Fig.5, it is not separated. To facilitate the reviewier’s reading , we added a box line around all the figures.

It is still unclear, what the practical and theoretical implications of the findings could be as they are very vague such as ""Further research and a larger sample size may be necessary to ascertain the full implications of SPP1 levels and their potential role in disease relapse or progression in the context of classical HL."

> Literature on pediatric Hodgkin lymphoma is limited due to the rarity of the disease and the low relapse rate. This manuscript has been requested by the editor for a special issue and we think that, altough preliminary, it is important to report our data because we observed a trend between increased osteopontin and worse patient outcomes. This data are important to understand the biological events accompagnying relapse, that have a great importance for clinicians dealing with the challenge to cure young patients that relapse. Our data holds on a comprehensive dataset, including pathological immunohistochemistry, laboratory and  PET imaging and molecular data from each patients.Further, there are very few papers including such a large series of different approach rearding patients within the same clinical trial. Our results are important as they show that osteopontin is not consistent enough to improve the standard, although not adequate, prognostic capacity of PET imaging, which is currently the best tool for predicting treatment response in children with HL in the clinic.

Finally, our results evidenced the importance of a specific subset of dendritic cells associated with increased osteopontin and the concomitant relapse of patients. This is reported for the first time in pediatric Hodgkin lymphoma, contributing to an overall deeper understanding of the biological hightroput photography of the complicate scenario that sustain HL development and relapse.

These informations have been further underlined in the last version of our paper, in the discussion section. However, although commonly used, we omitted the sentence indicated by the reviewer "Further research and a larger sample size may be necessary to ascertain the full implications of SPP1 levels and their potential role in disease relapse or progression in the context of classical HL”.

Extensive editing of English language and format required. Advice from an English native speaker is recommended.

> English has been reviewed, the certificate has been attached.